# Trends of attrition from HIV care and its predictors among adolescent girls and young women with inconsistent viral load suppression results in Mainland Tanzania, 2016–2024

Anthony Charles Kavindi[1]*, Jegede Feyisayo Ebeneezer[2],
Asteria Karungi Nyongoli[1], Paschal Yuda[1], Nagalal William[3],
Deogratius W. Kinoko[1], Laura J. Shirima[1], Festo Charles[4],
Marion Sumari-de Boer[1,3,5,6]

1 School of Public Health, KCMC University, Moshi, Tanzania, 2 Bayero University Kano, Department of Life Science, Kano, Nigeria, 3 Kilimanjaro Clinical Research Institute, Moshi, Tanzania, 4 Management and Development for Health (MDH), Dar es Salaam, Tanzania, 5 Amsterdam Institute for Global Health and Development, Amsterdam, the Netherlands, 6 Kilimanjaro Christian Medical Centre, Moshi, Tanzania

* kavindianthony@gmail.com

## Abstract

Adolescent Girls and Young Women (AGYW) in Tanzania Mainland are disproportionately affected by HIV. They are at high risk of attrition from HIV care, undermining efforts to achieve the UNAIDS 95-95-95 targets by 2030. AGYW with inconsistent viral load suppression are particularly vulnerable, leading to suboptimal treatment outcomes and continued transmission risks. This study aims to assess attrition in care and identify factors associated with persistent unsuppressed viral load after ART initiation among adolescent girls and young women in Tanzania Mainland from 2016 to 2024. This study utilized retrospective data from the National AIDS Control Programme (NACP) CTC database, covering the period from 2016 to 2024. Data were accessed for research purposes on 19/05/2025. Survival analysis techniques were employed to estimate attrition rates and identify predictors among AGYW with inconsistent viral load suppression results. The overall attrition rate was 11.8 per 1,000 person-years. Attrition was higher among AGYW aged 20–24 years (AHR: 1.58; 95% CI: 1.31–1.91; $p < 0.001$) compared to those aged 15–19 years, and among those residing in rural areas (AHR: 1.15; 95% CI: 1.05–1.26; $p = 0.002$). Participants with an initial viral load ≥1,000 cp/mL had an increased risk of attrition (AHR: 1.28; 95% CI: 1.16–1.42; $p < 0.001$), as did those attending public facilities (AHR: 1.79; 95% CI: 1.23–2.61; $p = 0.002$). Protective factors included second-line ART (AHR: 0.63; 95% CI: 0.55–0.73; $p < 0.001$), ART duration ≥4 years (AHR: 0.43; 95% CI: 0.29–0.64; $p < 0.001$), and residence in the Lake Zone (AHR: 0.64; 95% CI: 0.54–0.76; $p < 0.001$). Early attrition from HIV care is common among AGYW with inconsistent viral load suppression, particularly in the first year. Tailored interventions targeting

**Data availability statement:** The data used in this study are owned by the National HIV AIDS and STI Control Programme (NASHCOP), Ministry of Health, Tanzania. Due to ethical and legal restrictions, the dataset cannot be publicly shared. However, de-identified data may be made available upon reasonable request to NASCOP through the Ministry of Health, Tanzania (contact: info@nacp.go.tz) for researchers who meet the criteria for access to confidential data.

**Funding:** The authors received no specific funding for this work.

**Competing interests:** The authors have declared that no competing interests exist.

at-risk groups—based on age, residence, ART regimen, and facility type are urgently needed to improve retention and treatment outcomes in this population.

## Introduction

Human Immunodeficiency Virus (HIV) is a virus that weakens the immune system by targeting and destroying CD4 T cells, which play a crucial role in defending the body against infections. Without treatment, HIV progresses through four stages: acute infection, clinical latency, symptomatic HIV infection, and finally, acquired immunodeficiency syndrome (AIDS) [1].

Globally, HIV remains a major public health concern. As of 2024, approximately 39.9 million people were living with HIV [2]. Adolescent Girls and Young Women (AGYW) account for 1.9 million of these cases, and about 4,000 AGYW are newly infected each week. In 2023, 77% of all new HIV infections among AGYW occurred in Sub-Saharan Africa, with Eastern and Southern Africa contributing about 60% of new HIV infection cases.[2].

The high burden of new HIV infections among Adolescent Girls and Young Women (AGYW) contributed to global efforts to end the AIDS epidemic, leading UNAIDS to launch the 95-95-95 targets. These targets aim 95% of people living with HIV to know their status, 95% of those diagnosed to be on antiretroviral therapy (ART), and 95% of those on ART to achieve viral suppression by 2030 (UNAIDS, 2023). The UNAIDS targets focus on treating all PLHW and the same time prevention modality to reduce HIV transmission. Despite these efforts, 2023, global achievements still lagged: 86% of people living with HIV knew their status, 77% were on ART, and 72% were virally suppressed [3].

Effective HIV prevention strategies, including condom use, Pre-exposure Prophylaxis (PrEP), and Prevention of Mother-to-child Transmission (PMTCT), are essential to reducing new infections [4]. Moreover, achieving widespread ART coverage not only improves health outcomes for people living with HIV (PLHIV) but also serves as a key prevention strategy. When PLHIV achieve and maintain viral suppression (undetectable viral load), they cannot transmit HIV to others, a concept known as Treatment as Prevention (TasP), thereby reducing new infections [5]. Engagement of communities and stigma reduction efforts are also crucial to expanding access to ART for key populations, such as AGYW, sex workers, and men who have sex with men [1].

Despite progress, retention in HIV care remains a major challenge in Sub-Saharan Africa (SSA) among People living with HIV (PLHIV) receiving Antiretroviral Therapy (ART). A study by Haas et al. (2018) found that, 5 years after ART initiation, only 52.1% of PLHIV remained in care, with 41.8% lost to follow-up and 6.0% deceased [6]. After accounting for undocumented deaths and self-transfers, the ART retention estimate increased to 66.6%. Similarly, a meta-analysis of 32 ART programs in SSA reported decreasing retention rates over time: 80% after one year, 77% after two years, and 72% after three years, with loss to follow-up and death being the main contributors to attrition [7].

In Tanzania, the 2023 Tanzania HIV Impact Survey (THIS) reported an HIV prevalence of 4.5% among adults, with higher rates in women (5.6%) than in men (3.0%)

[8]. Among AGYW aged 20–24 years, the prevalence was 1.8%, double that of males in the same age group (0.9%). The HIV incidence among adults was 0.18%, while AGYW had a higher incidence of 0.33%, contributing 80% of new infections [8]. Adult PLHIV on ART showed a viral suppression rate of 94.3% (94.9% among women and 92.9% among men), the rate was notably lower among AGYW at 86.6%. Even more concerning, among all AGYW living with HIV (regardless of ART status), only 51.3% were virally suppressed, compared to 71.5% among young men [8]. These figures suggest that AGYW face heightened risks of disease progression and ongoing transmission due to inconsistent viral suppression and retention in care.

Understanding the patterns and predictors of attrition from HIV care among AGYW with unsuppressed or inconsistent viral load is therefore critical to achieving HIV epidemic control.

This study examined trends in ART care attrition, survival time in care, and associated factors among AGYW in Tanzania Mainland.

## Methods

### Study design and data source

We conducted a retrospective cohort study using routinely collected, de-identified data from people living with HIV (PLHIV) attending Care and Treatment Clinics (CTCs) across Mainland Tanzania from 2016 to 2024. Healthcare providers from 1,135 facilities (487 dispensaries, 432 health centers, and 216 hospitals) recorded socio-demographic, clinical, laboratory, and pharmacological information during patient visits.

### Study population

Adolescent girls and young women (AGYW) aged 15–24 years were eligible if they meet the following criteria

1. Were aged 15–24 years during the study enrollment period time (2016–2019)

2. Had been on ART for at least six months.

3. Had documented inconsistent viral load suppression, defined as PLHIV who initially achieved viral suppression (undetectable viral load ≥6 months after ART initiation) but subsequently experienced one or more detectable viral load results ≥1000 copies/ml during follow-up; and

Eligible participants were followed longitudinally from their initial enrollment until attrition from HIV care or the end of the study period in 2024, whichever occurred first. The timeframe 2016–2024 refers to the full observation period during which participants' initial visits, follow-up, and outcome assessments were captured in the dataset.

### Data source and data extraction

Data were extracted from the CTC2 database on 19/05/2025, which links patient-held CTC1 cards and facility-based CTC2 cards, supported by pre-ART and ART registers and cohort analysis reports. Unique patient identification numbers ensured consistent longitudinal tracking. We excluded patients who transferred without follow-up or lacked viral load tests. From over 2.6 million PLHIV, we identified 7,910 eligible AGYW for the final analysis.

### Study variables

**Dependent variable.** The dependent variable in this study was attrition from HIV care. It was defined as a binary outcome: Yes [1] for PLHIV who had ever experienced attrition (i.e., were not in care at any point during the follow-up period) and No (0) for those who remained continuously engaged in HIV care throughout the follow-up period.

**Independent variables.** Independent (predictor) variables were categorized into socio-demographic, clinical, and health system factors.

Key socio-demographic, clinical, and facility-level variables were included in the analysis. Age was measured at ART initiation and study enrollment and categorized into 15–19 and 20–24 years. Residence was classified as urban or rural, while marital status was recorded as single, married, or divorced/separated. Facility-related variables included type/level of health facility (dispensary, health center, hospital) and ownership (government, private, or faith-based). Geographical zone was categorized based on regions of Mainland Tanzania. Clinical variables included entry point at HIV diagnosis, initial viral load after six months on ART, baseline CD4 + cell count, WHO HIV clinical stage at ART initiation, and duration on ART. Detailed coding and operational definitions for each variable are provided in S1 Table.

### Data analysis plan

Analyses were performed using Stata version 18.0 (Stata Corp LLC, College Station, TX, USA). Descriptive statistics summarized participants' sociodemographic and clinical characteristics. Continuous variables (e.g., age at enrollment, age at ART initiation, and duration on ART) were summarized using medians and interquartile ranges (IQRs). In contrast, categorical variables were summarized using frequencies and proportions.

To address the study objectives, we conducted the following analyses.

**Objective 1** was to determine the proportion of AGYW with attrition from HIV care over five years; for this, we prepared time-to-event data and estimated attrition rates per 1,000 person-years for each follow-up year.

**Objective 2** aimed to assess survival time in care among AGYW; Kaplan–Meier survival analysis was used to estimate median survival time and cumulative retention probabilities. Time-to-event was defined as the time from ART initiation to attrition (loss to follow-up, treatment cessation, or death), and survival curves were stratified by marital status, duration on ART, ART regimen, and initial viral load, with differences evaluated using the log-rank test.

**Objective 3** sought to identify predictors of attrition; a Weibull regression model with gamma-distributed shared frailty at the individual (CTC ID) level was fitted. Univariable models were first performed, and variables with likelihood-ratio test p-values <0.2 were included in the multivariable model. Multicollinearity was assessed using the Variance Inflation Factor (VIF), model fit via likelihood ratio tests, and potential confounding was evaluated by comparing crude and adjusted hazard ratios, with a ≥ 15% change considered indicative of confounding.

## Results

### Study participants characteristics

A total of 7,910 AGYW were analyzed. The median age at ART initiation was 17 years (IQR: 9–18), with 65.0% starting before age 15. At enrollment, the median age was 17 years (IQR: 15–21), and 66.7% were aged 15–19 years. Among those with known marital status (n = 6,256), 76.4% were single and 22.2% married. Most received care at public facilities (71.6%) and hospitals (53.4%), while 25.5% received care at faith-based facilities. HIV was mainly diagnosed at VCT (51.4%), followed by OPD/IPD (25.0%) and PMTCT/RCH (13.8%) (n = 7,141). Most participants were from the Southern Highlands (30.6%) and Coastal Zones (26.9%), with 51.7% in urban areas. First-line ART regimens were used by 90.4%, with a median duration of 105 months (IQR: 83–141). At ART initiation, 30.0% had CD4 < 200 cells/mm³, and 32.9% had CD4 ≥ 500 cells/mm³. Over half (52.3%) were WHO Stage 3, and initial viral load showed 56.4% were unsuppressed (≥1000 copies/mL) and 23.9% undetectable (<50 copies/mL) (See Table 1).

### Overall attrition rate from HIV care

A total of 5,100 attrition events occurred over 432,573.7 person-years (PY) of follow-up, yielding an overall attrition rate of 11.8 per 1,000 PY (95% CI: 11.15–11.21). This implies that for every 1,000 AGYW followed for one year, approximately 11.8 were not retained in care.

**Table 1. Study demographic and clinical characteristics of the study AGYW (n = 7,910).**

| Variable | Frequency (n = 7,910) | Percentage (%) |
|---|---|---|
| **Age at start ART (Years)** | | |
| -14 | 5,142 | 65.0 |
| 15-19 | 1,460 | 18.5 |
| 20-24 | 1,308 | 16.5 |
| *Median (IQR)* | 17(9,18) | |
| **Age at study enrollment (Years)** | | |
| 15-19 | 5,274 | 66.7 |
| 20-24 | 2,636 | 33.3 |
| *Median (IQR)* | 17(15,21) | |
| **Marital Status (n = 6,256)*** | | |
| Single | 4,780 | 76.4 |
| Married | 1,386 | 22.2 |
| Divorced | 90 | 1.4 |
| **Facility ownership** | | |
| Private | 234 | 3.0 |
| Public | 5,662 | 71.6 |
| FBO | 2,014 | 25.5 |
| **Facility type** | | |
| Dispensary | 1,297 | 16.4 |
| Health Centre | 2,285 | 28.9 |
| Hospital | 4,225 | 53.4 |
| Other | 103 | 1.3 |
| **Entry point at HIV diagnosis (n = 7,141)*** | | |
| OPD/IPD | 1,786 | 25.0 |
| RCH/PMTCT | 988 | 13.8 |
| VCT | 3,673 | 51.4 |
| Community | 124 | 1.7 |
| Others | 570 | 8.0 |
| **Zone** | | |
| Central Zone | 550 | 7.0 |
| Coastal Zone | 2,129 | 26.9 |
| Lake Zone | 1,371 | 17.3 |
| Northern Zone | 1,247 | 15.8 |
| Southern Highland Zone | 2,420 | 30.6 |
| Western Zone | 193 | 2.4 |
| **Residence** | | |
| Urban | 4,091 | 51.7 |
| Rural | 3,819 | 48.3 |
| **ART regimen** | | |
| First Line | 7,148 | 90.4 |
| Second line | 762 | 9.6 |
| **CD4 + count -at baseline (n = 6,868)* (cells/mm³)** | | |
| <200 | 2,063 | 30.0 |
| 200 - 349 | 1,426 | 20.8 |
| 350 - 499 | 1,117 | 16.3 |
| 500+ | 2,262 | 32.9 |

*(Continued)*

**Table 1.** (Continued)

| Variable | Frequency (n = 7,910) | Percentage (%) |
|---|---|---|
| **WHO stage (n = 7,578)\*** | | |
| Stage 1 | 1,222 | 16.1 |
| Stage 2 | 1,277 | 16.9 |
| Stage 3 | 3,966 | 52.3 |
| Stage 4 | 1,113 | 14.7 |
| **Duration on ART (Years)** | | |
| <3 | 89 | 1.1 |
| 4+ | 7,821 | 98.9 |
| *Median (IQR)* | 105(83,141) | |
| **Initial VL result (copies/mL)** | | |
| <50 undetectable | 1,889 | 23.9 |
| 50–999 LLV | 1,556 | 19.7 |
| >=1000 unsuppressed | 4,465 | 56.4 |

Key: * Frequencies (n) do not tally to the total due to the missing value; OPD : Outpatient Department VCT : voluntary Counseling and Testing, IPD : Inpatient Department, RCH : Reproductive and Child Health, LLV : Low-Level Viremia, ART First Line = Tenofovir + Lamivudine + Dolutegravir , ART Second line = Tenofovir + Lamivudine + Efavirenz

## Five-year trends in attrition rates

Attrition was highest during the first year, at 15.8 per 1,000 person-years (95% CI: 15.1–16.7), and gradually decreased over the follow-up period: 13.1 per 1,000 person-years (95% CI: 12.4–13.1) in year 2, 12.1 per 1,000 person-years (95% CI: 11.4–12.9) in year 3, 8.9 per 1,000 person-years (95% CI: 8.3–9.6) in year 4, and 7.4 per 1,000 person-years (95% CI: 6.8–8.1) in year 5 (Fig 1).

## Attrition rate from HIV care among AGYW by geographical zones

The analysis revealed notable variations in attrition rates from HIV care among Adolescent Girls and Young Women (AGYW) with inconsistent viral load suppression across geographical zones in Mainland Tanzania Fig 2. As illustrated in Fig 4, the Central Zone exhibited the highest attrition rate, with 14.78 per 1,000 person-years (PY) (95% CI: 13.46–16.22), indicating a comparatively elevated risk of disengagement from care. This was followed by the Northern Zone (12.91 per 1,000 PY), Southern Highland Zone (12.42 per 1,000 PY), and Western Zone (11.48 per 1,000 PY). Conversely, the Coastal Zone and Lake Zone demonstrated the lowest attrition rates, at 10.57 per 1,000 PY (95% CI: 10.00–11.19) and 10.41 per 1,000 PY (95% CI: 9.71–11.15), respectively, suggesting relatively stronger retention in care within these zones.

Fig 3 presents the attrition rates from HIV care among adolescent girls and young women (AGYW) with inconsistent viral load suppression across regions in Mainland Tanzania, expressed per 1,000 person-years (PY) with corresponding 95% confidence intervals. Singida Region recorded the highest attrition rate at 22.01 per 1,000 PY (95% CI: 18.54–26.12), reflecting a significantly elevated risk of disengagement from care. Other regions with relatively high attrition rates included Manyara, Kilimanjaro, and Morogoro, each exceeding 14 per 1,000 PY. In contrast, Dar es Salaam, Mwanza, Kagera, and Katavi reported the lowest attrition rates, all below 10 per 1,000 PY, with Dar es Salaam demonstrating the most precise estimate (95% CI: 8.41–9.84), indicating better retention in HIV care within the region.

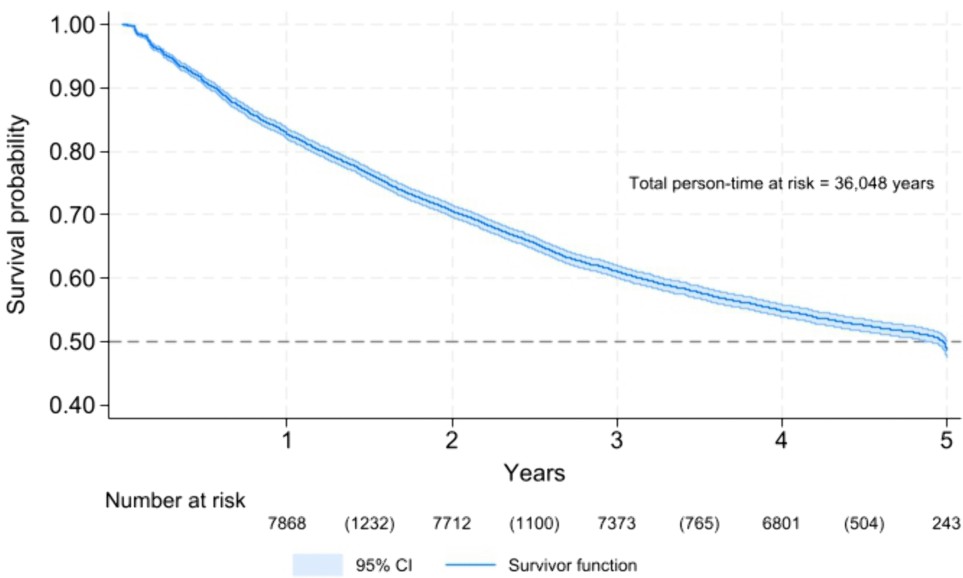

**Fig 1. Probability of attrition from HIV care among the AGWY with inconsistent viral load suppression over time suppression.**

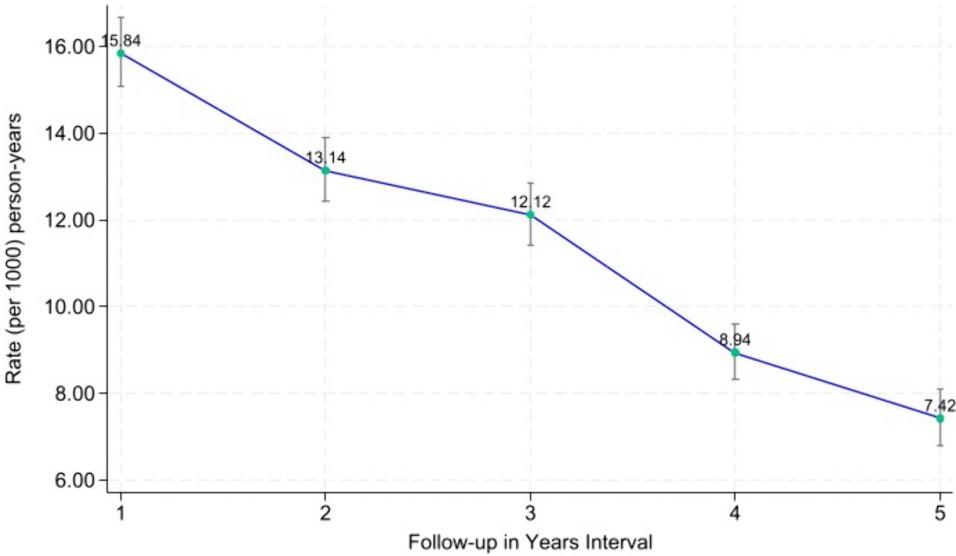

**Fig 2. Five-year trend in attrition rates from HIV care per 1000 years.**

## Survival time in HIV care

Survival analysis over the five-year follow-up period showed a progressive decline in retention among adolescent girls and young women (AGYW), with 82.7% remaining in care at the end of the first year, decreasing to 70.6% in year two, 61.1% in year three, 54.9% in year four, and 48.7% by year five. These findings indicate that nearly half of the cohort

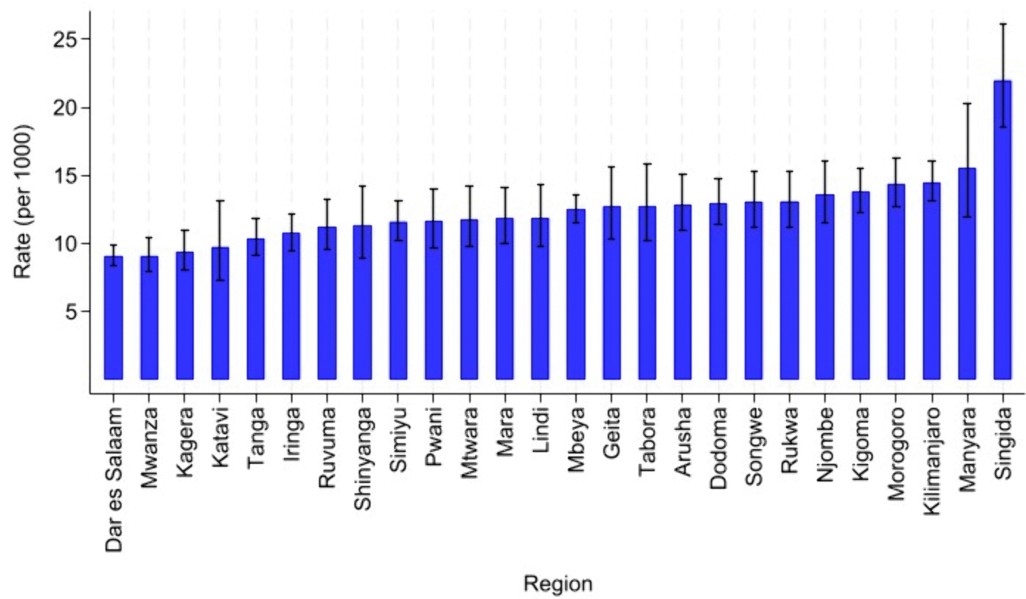

**Fig 3. Attrition rate from HIV care among AGYW by Region per 1000 years.**

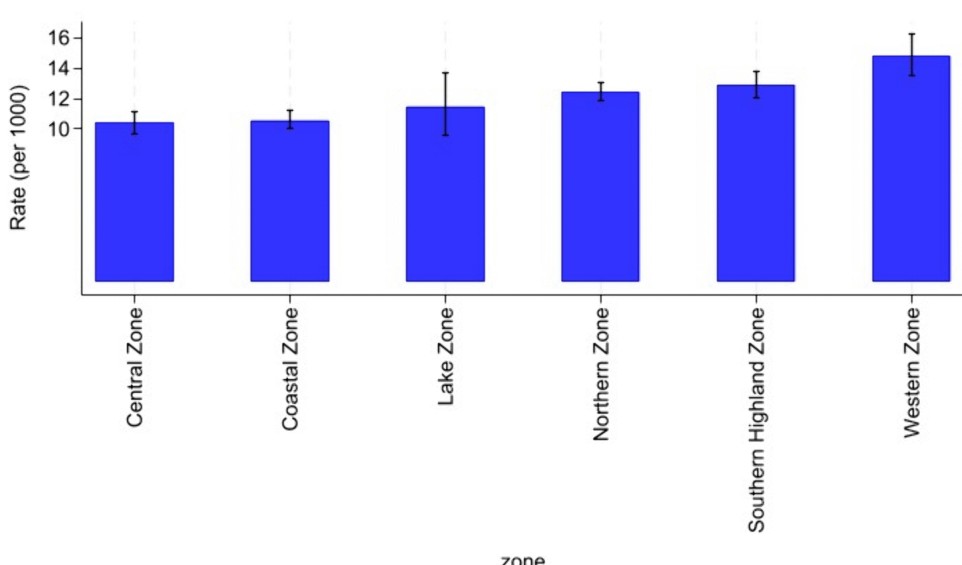

**Fig 4. Attrition rate from HIV care among AGYW by geographical zones per 1000 years.**

experienced attrition within five years follow-up period. Overall, there was a consistent decline in the probability of remaining in care over time. The median survival time, defined as the time by which 50% of the participants were still engaged in care, was reached between the fourth and fifth year, suggesting that half of the AGYW cohort remained in HIV care for approximately 4–5 years following ART initiation (Fig 1).

## Sociodemographic and clinical predictors of attrition from HIV care

A multivariable Weibull proportional hazards regression model with gamma shared frailty was used to identify predictors of attrition from HIV care among AGYW with inconsistent viral load suppression in mainland Tanzania (2016–2024) Fig 5. Individual-level frailty accounted for unobserved heterogeneity ($\theta = 0.50$; 95% CI: 0.41–0.62; $p < 0.001$). The Weibull shape parameter was $< 1$ ($p = 0.87$; 95% CI: 0.84–0.90), indicating a declining hazard over time and higher attrition early in care.

After adjustment, adolescents aged 15–19 and young women aged 20–24 at ART initiation had higher attrition risk than those aged 0–14 (AHR = 1.50 and 1.58; $p < 0.001$). Conversely, those aged 20–24 at enrollment were less likely to disengage than those aged 15–19 years (AHR = 0.85; $p = 0.028$). Receiving care from public or faith-based facilities increased attrition risk compared to private facilities (AHR = 1.79 and 1.73), and clients in "Other" facility types had nearly threefold higher risk (AHR = 2.68; $p < 0.001$).

Regional differences were evident: clients from Coastal, Lake, and Southern Highland Zones had lower attrition than those in the Central Zone. Rural residence increased attrition risk (AHR = 1.15; $p = 0.002$). Participants with initial viral load ≥1000 copies/mL had 28% higher attrition (AHR = 1.28; $p < 0.001$). Those on second-line ART (AHR = 0.63; $p < 0.001$) and those on ART for ≥4 years (AHR = 0.43; $p < 0.001$) were less likely to disengage (See Table 2).

## Discussion

This retrospective cohort study assessed five-year trends and predictors of attrition from HIV care among AGYW with inconsistent viral load suppression in mainland Tanzania (2016–2024), using routine CTC data. The overall attrition rate was 11.8 per 1,000 person-years (PY), aligning with findings from Namibia [9] and sub-Saharan Africa [10], but lower than rates reported in Kenya, Cameroon [11,12], and Mozambique [13]. These differences are multifaceted ranging from variation in studied populations as some studies included broader age groups or newly initiated ART clients. Additionally, other differences may be associated with nature of programmatic support including clients tracking systems, health system capacity, definitions of attrition and follow-up duration across settings. The data showed finding from this work revealed

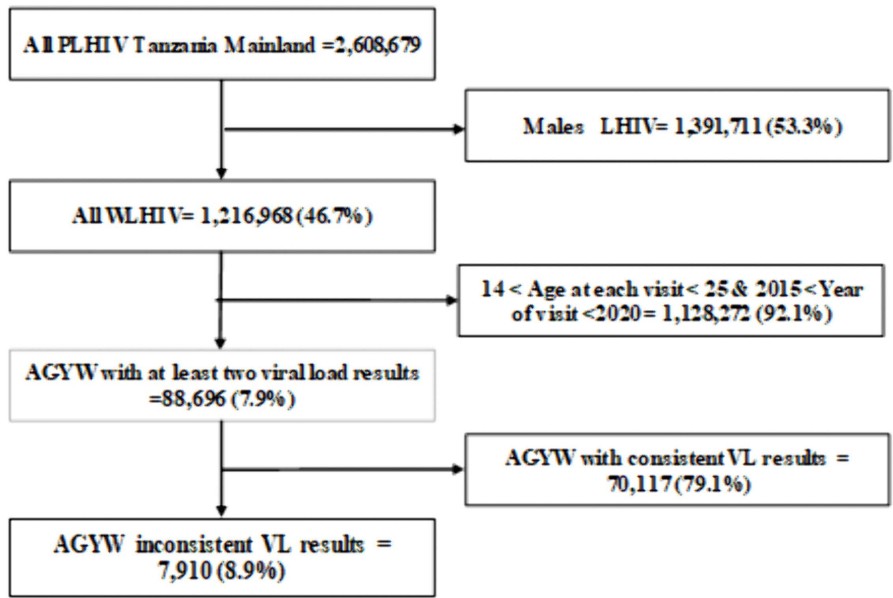

**Fig 5. Flow chart showing the recruitment of the AGYW with inconsistent viral load suppression.**

**Table 2. Multivariable analysis for social demographic and clinical predictors of attrition from HIV care among AGYW with inconsistent viral load suppression in Mainland Tanzania from 2016 to 2024 (N = 5,093).**

| Variable | CHR (95% CI) | p-value | AHR (95% CI) | p-value |
|---|---|---|---|---|
| **Age at Start ART (Years)** | | | | |
| 0-14 | 1 | | 1 | |
| 15-19 | 1.37 (1.26-1.48) | < 0.001 | 1.50 (1.31-1.73) | < 0.001 |
| 20-24 | 1.47 (1.35-1.59) | < 0.001 | 1.58 (1.31-1.91) | < 0.001 |
| **Age – at enrollment (Years)** | | | | |
| 15-19 | 1 | | 1 | |
| 20-24 | 1.27 (1.19-1.36) | < 0.001 | 0.85 (0.73-0.98) | 0.028 |
| **Marital Status** | | | | |
| Single | 1 | | 1 | |
| Married | 1.22 (1.12-1.33) | < 0.001 | 0.99 (0.88-1.12) | 0.915 |
| Divorced | 1.31 (0.98-1.74) | 0.067 | 1.03 (0.75-1.42) | 0.843 |
| **Facility Ownership** | | | | |
| Private | 1 | | 1 | |
| Public | 1.21 (0.98-1.48) | 0.070 | 1.79 (1.23-2.61) | 0.002 |
| FBO | 1.15 (0.93-1.42) | 0.208 | 1.73 (1.18-2.52) | 0.005 |
| **Facility Type** | | | | |
| Dispensary | 1 | | 1 | |
| Health Centre | 1.04 (0.94-1.14) | 0.461 | 1.11 (0.96-1.27) | 0.153 |
| Hospital | 0.91 (0.83-0.99) | 0.034 | 1.02 (0.89-1.16) | 0.817 |
| Other | 1.08 (0.81-1.44) | 0.606 | 2.68 (1.57-4.59) | 0.000 |
| **Entry point at HIV diagnosis** | | | | |
| OPD/IPD | 1 | | 1 | |
| RCH/PMTCT | 1.23 (1.1-1.37) | < 0.001 | 0.97 (0.84-1.13) | 0.731 |
| VCT | 0.91 (0.83-0.99) | 0.021 | 0.93 (0.84-1.03) | 0.154 |
| Community | 0.78 (0.59-1.04) | 0.092 | 0.81 (0.56-1.17) | 0.253 |
| Others | 0.92 (0.8-1.06) | 0.249 | 0.92 (0.77-1.11) | 0.392 |
| **Zone** | | | | |
| Central Zone | 1 | | 1 | |
| Coastal Zone | 0.72 (0.63-0.82) | < 0.001 | 0.73 (0.61-0.87) | < 0.001 |
| Lake Zone | 0.71 (0.61-0.81) | < 0.001 | 0.64 (0.54-0.76) | < 0.001 |
| Northern Zone | 0.87 (0.76-1.00) | 0.054 | 0.89 (0.75-1.06) | 0.199 |
| Southern Highland Zone | 0.84 (0.74-0.96) | 0.008 | 0.78 (0.66-0.92) | 0.003 |
| Western Zone | 0.78 (0.61-0.98) | 0.036 | 0.81 (0.60-1.09) | 0.160 |
| **Residence** | | | | |
| Urban | 1 | | 1 | |
| Rural | 1.17 (1.1-1.25) | < 0.001 | 1.15 (1.05-1.26) | 0.002 |
| **CD4 + Count (cells/mm³)** | | | | |
| <200 | 1 | | 1 | |
| 200 – 349 | 1.04 (0.94-1.15) | 0.478 | 0.96 (0.85-1.08) | 0.497 |
| 350 – 499 | 1.1 (0.99-1.22) | 0.089 | 1.07 (0.95-1.22) | 0.263 |
| 500+ | 0.98 (0.9-1.07) | 0.665 | 0.95 (0.85-1.06) | 0.354 |
| **Initial VL Test (copies/mL)** | | | | |
| <50 undetectable | 1 | | 1 | |
| 50 – 999 | 1.13 (1.02-1.25) | 0.017 | 1.13 (0.99-1.28) | 0.070 |
| ≥1000 | 1.19 (1.1-1.29) | < 0.001 | 1.28 (1.16-1.42) | < 0.001 |

*(Continued)*

**Table 2.** (Continued)

| Variable | CHR (95% CI) | p-value | AHR (95% CI) | p-value |
|---|---|---|---|---|
| **ART Group** | | | | |
| First Line | 1 | | 1 | |
| Second Line | 0.69 (0.62-0.76) | < 0.001 | 0.63 (0.55-0.73) | < 0.001 |
| **Time on ART (Years)** | | | | |
| <4 | 1 | | 1 | |
| ≥4 | 0.42 (0.3-0.58) | < 0.001 | 0.43 (0.29-0.64) | < 0.001 |

Key: AIC = 15869.75; BIC= 16173.92; p = 0.87; θ = 0.50; CHR = Crude Hazards Ratio: AHR= Adjusted Hazards Ratio, ART First Line = Tenofovir + Lamivudine + Dolutegravir, ART Second line = Tenofovir + Lamivudine + Efavirenz

Abbreviations: OPD: Outpatient Department VCT: voluntary Counseling and Testing, IPD: Inpatient Department, RCH: Reproductive and Child Health, LLV: Low-Level Viremia.

early attrition from HIV care which was highest during the first year of follow-up, consistent with studies from South Africa [14], highlighting initial treatment challenges and psychosocial stressors.

Attrition from HIV care was notably higher among AGYW aged 20–24 compared to younger age groups, echoing regional evidence that transitions to adult responsibilities contribute to disengagement. Those with initial unsuppressed viral loads had a higher risk of attrition, supporting WHO findings that poor viral suppression predicts future disengagement [14].

Attrition from HIV care declined steadily over the five years but remained substantial, underscoring the need for adolescent-specific interventions such as peer support, mobile health (mHealth) strategies, and community outreach. Data evidence suggests that, marital status influenced HIV care retention, with married and divorced AGYW experiencing higher attrition than their single peers. This association may reflect competing domestic responsibilities, reduced autonomy in healthcare decision-making, or relationship-related stigma that can affect consistent engagement in care. However, findings on the role of marital status vary across settings; in some contexts, marriage has been associated with improved social support and better retention in care. The observed association in this study may also be partly explained by the cohort composition, which included a smaller proportion of married AGYW (22%) compared to single individuals (76%) and a higher representation of younger women, potentially influencing patterns of care engagement [15,16].

Clients on first-line ART experienced higher attrition than those on second-line regimens. Although enhanced monitoring of second-line clients has been reported in other settings, such as Rwanda [17], Tanzanian national guidelines do not explicitly differentiate adherence counselling or follow-up intensity by ART regimens. The observed difference in this study may therefore reflect closer clinical attention triggered by treatment failure, increased patient awareness of disease severity, or selection effects among clients who remain engaged long enough to transition to second-line therapy, rather than a formal programmatic distinction in care provision. Attrition was lower among AGYW with advanced WHO clinical stages, possibly reflecting closer clinical follow-up for those with more severe disease. This finding contrasts with evidence from adult-dominated cohorts, such as a rural Mozambique study where higher WHO clinical stages were associated with increased attrition; however, only a small proportion (14.2%) of participants in that study were aged 15–24 years, limiting direct comparability with our AGYW-focused cohort [13].

Younger age at ART initiation (<15 years) was protective, while older initiation ages were associated with a higher hazard of attrition, consistent with prior studies in Uganda [15]. Higher attrition was also observed among AGYW with initial viral loads ≥1,000 copies/mL and among those residing in rural areas, where geographic and structural barriers may limit consistent access to care. These findings reinforce the importance of tailoring HIV services to adolescent needs, especially for rural and high-risk populations [15,18].

## Strengths and limitations of the study

This study has several limitations. Classifying individuals lost to follow-up as attrition may have led to an overestimation of attrition rates, as some clients might have continued care at other facilities under different CTC identification numbers. Additionally, the dataset lacked key behavioral, social, and structural variables such as stigma, mental health status, and household support, which are important predictors of attrition. The use of routine programmatic data collected for service delivery rather than research may also have introduced issues such as incomplete records, misclassification of outcomes, and inconsistencies in data due to high workloads and limited data validation at the facility level. Despite these limitations, the study has notable strengths. It used a large, nationally representative dataset from the CTC2 database, covering all regions and facilities offering HIV services in Tanzania. The long five-year follow-up period and large sample size increased statistical power. At the same time, the use of real-world data enhances the relevance of findings for policy and program decision-making. The focus on adolescent girls and young women a population disproportionately affected by HIV further adds value to the study by identifying critical gaps in care for this vulnerable group.

## Recommendation

To address the high attrition from HIV care among AGYW with inconsistent viral load suppression, an integrated approach combining facility and community-level interventions is essential.

Key strategies include enhancing support in the first year of ART through structured peer mentorship, improved adherence counseling, and the use of digital health tools to strengthen retention and follow-up. For instance, Jichunge App [19] which sends automated SMS reminders for clinic appointments and medication adherence,. Youth-friendly service models should be reinforced through adolescent-focused clinic days, extended service hours, and digital health interventions. Mobile health (mHealth) platforms, such as SMS reminders, telehealth consultations, and mobile apps like T-HIT [20], which can support adherence, retention, and timely follow-up, although their national coverage remains limited.

In rural areas where attrition is more pronounced efforts should emphasize community ART groups, outreach services, and multi-month dispensing. For clients with initially high viral loads, timely monitoring, intensified adherence counseling, and prompt regimen changes are crucial. Strengthened follow-up systems, particularly for AGYW enrolled through PMTCT and OPD, and involvement of community health workers for home visits and tracing, are also recommended. Lastly, addressing social and marital barriers through disclosure-supportive counseling and partner engagement can improve long-term retention.

## Conclusion

This study found that the overall attrition rate from HIV care among adolescent girls and young women (AGYW) with inconsistent viral load suppression was highest during the first year of follow-up. Notably, individuals with an initial unsuppressed viral load were more likely to disengage from care. Furthermore, the median survival time in HIV care progressively declined over the study period, with the first-year post-enrollment on HIV care emerging as the most critical period for retention. These findings highlight the urgent need to strengthen early Antiretroviral Therapy (ART) support during the first year of care, a period marked by the highest risk of disengagement. Attrition was more common among older AGYW (20–24 years) compared to those on first-line ART, residents of rural areas, individuals with high initial viral load (≥1,000 copies/mL), and those diagnosed through RCH/PMTCT or OPD services. Collectively, these results identify subgroups at heightened risk of attrition and provide an evidence base to inform the design and prioritization of differentiated HIV care approaches for AGYW in Tanzania.

## Ethical considerations

Ethical approval was obtained from the KCMC University Research Ethics Review Committee (CRERC) (Approval ID: PG195/2024, dated 7 March 2025). Data access permission was granted by the National AIDS and STI Control

Programme (NASCOP), Ministry of Health, Tanzania (Reference No. PA.104/262/01/51, dated 26 May 2025). The study used anonymized secondary data; thus, informed consent was not required. Confidentiality was maintained by excluding all personal identifiers from the analysis and reporting.

## Supporting information

**S1 Table. Categorization and coding of independent variables.**
(DOCX)

## Acknowledgments

We sincerely thank the National AIDS & STI Control Programme (NASCOP) and the Ministry of Health, Tanzania, for granting access to the CTC2 database, which made this study possible.

## Author contributions

**Conceptualization:** Anthony Charles Kavindi, Jegede Feyisayo Ebeneezer, Asteria Karungi Nyongoli, Nagalal William, Paschal Yuda, Deogratius W. Kinoko, Festo Charles, Marion Sumari-de Boer.

**Data curation:** Anthony Charles Kavindi.

**Formal analysis:** Anthony Charles Kavindi, Festo Charles.

**Methodology:** Anthony Charles Kavindi, Jegede Feyisayo Ebeneezer, Asteria Karungi Nyongoli, Laura J Shirima, Festo Charles, Marion Sumari-de Boer.

**Supervision:** Marion Sumari-de Boer.

**Visualization:** Anthony Charles Kavindi, Jegede Feyisayo Ebeneezer, Nagalal William, Paschal Yuda, Marion Sumari-de Boer.

**Writing – original draft:** Anthony Charles Kavindi, Jegede Feyisayo Ebeneezer, Nagalal William, Marion Sumari-de Boer.

**Writing – review & editing:** Anthony Charles Kavindi, Jegede Feyisayo Ebeneezer, Marion Sumari-de Boer.

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
