## [Decision Letter · Decision Letter 0]

10 Dec 2025

PGPH-D-25-03362

Trends of attrition from HIV care and its predictors among Adolescent Girls and Young Women with inconsistent viral load suppression results in Mainland Tanzania, 2016–2024.

Dear Dr. Kavindi,

Thank you for submitting your manuscript to PLOS Global Public Health. After careful consideration, we feel that it has merit but does not fully meet PLOS Global Public Health’s publication criteria as it currently stands. Therefore, we invite you to submit a revised version of the manuscript that addresses the points raised during the review process.

We look forward to receiving your revised manuscript.

Kind regards,

Joel Msafiri Francis, MD, MS, PhD

Academic Editor

Journal Requirements:

Additional Editor Comments (if provided):

Reviewers' comments:

Reviewer's Responses to Questions

**Comments to the Author**

1. Does this manuscript meet PLOS Global Public Health’s publication criteria?

Reviewer #1: Yes

Reviewer #2: Yes

2. Has the statistical analysis been performed appropriately and rigorously?

Reviewer #1: Yes

Reviewer #2: Yes

3. Have the authors made all data underlying the findings in their manuscript fully available (please refer to the Data Availability Statement at the start of the manuscript PDF file)?

Reviewer #1: Yes

Reviewer #2: Yes

4. Is the manuscript presented in an intelligible fashion and written in standard English?

Reviewer #1: Yes

Reviewer #2: Yes

Reviewer #1: Methodology:

study design & datasource, heading overlap with data source and extraction. You may consider differentiating the study design+population from the data source+data extraction to improve clarity and avoid repetition.

The covariates are presented in a table. Consider adding a concise paragraph describing key variables and their categorization. Perhaps detailed coding and operational definitions should be presented in a supplementary table.

The analysis plan refers to Objective 2 and Objective 3 however, the study objectives are not stated anywhere earlier in the manuscript. They should be explicitly stated to improve coherence

Also, this contradicts “shared frailty at the individual level”. Please clarify the intended clustering structure. Did you model with the shared or Individual-level frailty?

Reviewer #2: This manuscript addresses an important and under-researched area by examining long-term trends in attrition from HIV care and associated predictors among adolescent girls and young women (AGYW) with non-viral load suppression in Tanzania. The use of routine programmatic data is a major strength and enhances the relevance of the findings for the national program. The objectives are clear, and the results provide insights into patterns of disengagement from care in this vulnerable population. However, the manuscript would benefit from clearer operational definitions of key outcomes particularly inconsistent viral load suppression. Further discussion linking the findings to existing regional and global literature, as well as clearer description of programmatic and policy implications, would strengthen the paper. I recommend minor revisions to improve clarity and consistency.

**Do you want your identity to be public for this peer review?** For information about this choice, including consent withdrawal, please see our Privacy Policy

Reviewer #1: No

Reviewer #2: **Yes: ** Joan Rugemalila

---

## [Decision Letter · Decision Letter 1]

13 Jan 2026

PGPH-D-25-03362R1

Trends of attrition from HIV care and its predictors among Adolescent Girls and Young Women with inconsistent viral load suppression results in Mainland Tanzania, 2016–2024.

Dear Dr. Kavindi,

Thank you for submitting your manuscript to PLOS Global Public Health. After careful consideration, we feel that it has merit but does not fully meet PLOS Global Public Health’s publication criteria as it currently stands. Therefore, we invite you to submit a revised version of the manuscript that addresses the points raised during the review process.

We look forward to receiving your revised manuscript.

Kind regards,

Joel Msafiri Francis, MD, MS, PhD

Academic Editor

Journal Requirements:

1. Please amend your online Financial Disclosure statement. If you did not receive any funding for this study, please simply state: “The authors received no specific funding for this work.”

2. Please update your online Competing Interests statement. If you have no competing interests to declare, please state: “The authors have declared that no competing interests exist.”

3. Please ensure that you cite or refer to Figures 2 and 5 in your text as, if accepted, production will need these references to link the reader to the figures.

4. "Supplementary table.docx" is currently uploaded as an 'Other' file type, which is not viewable by reviewers. Please ensure that all files meant for review are uploaded as 'Supporting Information' and include a legend in the manuscript.

Additional Editor Comments (if provided):

Reviewers' comments:

Reviewer's Responses to Questions

**Comments to the Author**

Reviewer #1: All comments have been addressed

Reviewer #2: All comments have been addressed

publication criteria?

Reviewer #1: Yes

Reviewer #2: Yes

3. Has the statistical analysis been performed appropriately and rigorously?

Reviewer #1: Yes

Reviewer #2: Yes

4. Have the authors made all data underlying the findings in their manuscript fully available (please refer to the Data Availability Statement at the start of the manuscript PDF file)?

Reviewer #1: Yes

Reviewer #2: Yes

5. Is the manuscript presented in an intelligible fashion and written in standard English?

Reviewer #1: Yes

Reviewer #2: Yes

Reviewer #1: (No Response)

Reviewer #2: Well addressed and improved revision; however, a few minor edits are recommended.

**Do you want your identity to be public for this peer review?** For information about this choice, including consent withdrawal, please see our Privacy Policy

Reviewer #1: **Yes: ** Muhere Cresensia Felician

Reviewer #2: No

---

## [Editor Report · Decision Letter 2]

22 Jan 2026

Trends of attrition from HIV care and its predictors among Adolescent Girls and Young Women with inconsistent viral load suppression results in Mainland Tanzania, 2016–2024.

PGPH-D-25-03362R2

Dear Kavindi,

We are pleased to inform you that your manuscript 'Trends of attrition from HIV care and its predictors among Adolescent Girls and Young Women with inconsistent viral load suppression results in Mainland Tanzania, 2016–2024.' has been provisionally accepted for publication in PLOS Global Public Health.

Best regards,

Joel Msafiri Francis, MD, MS, PhD

Academic Editor